# A Lagrangian Duality Approach to Active Learning

**Juan Elenter** *
University of Pennsylvania
elenter@seas.upenn.edu

**Navid NaderiAlizadeh**
University of Pennsylvania
nnaderi@seas.upenn.edu

**Alejandro Ribeiro**
University of Pennsylvania
aribeiro@seas.upenn.edu

## Abstract

We consider the pool-based active learning problem, where only a subset of the training data is labeled, and the goal is to query a batch of unlabeled samples to be labeled so as to maximally improve model performance. We formulate the problem using constrained learning, where a set of constraints bounds the performance of the model on labeled samples. Considering a primal-dual approach, we optimize the primal variables, corresponding to the model parameters, as well as the dual variables, corresponding to the constraints. As each dual variable indicates how significantly the perturbation of the respective constraint affects the optimal value of the objective function, we use it as a proxy of the informativeness of the corresponding training sample. Our approach, which we refer to as Active Learning via Lagrangian dualitY, or ALLY, leverages this fact to select a diverse set of unlabeled samples with the highest estimated dual variables as our query set. We demonstrate the benefits of our approach in a variety of classification and regression tasks and discuss its limitations depending on the capacity of the model used and the degree of redundancy in the dataset. We also examine the impact of the distribution shift induced by active sampling and show that ALLY can be used in a generative mode to create novel, maximally-informative samples.

## 1 Introduction

One of the key drivers of the recent progress in machine learning is the availability of massive, high-quality datasets, which enables training models comprising millions, or even billions, of parameters [1, 2, 3]. Nevertheless, in some areas, such as healthcare, obtaining *labeled* training data is challenging and/or expensive [4, 5, 6]. This has given rise to a class of approaches, collectively referred to as *active learning*, whose goal is to minimize the labeling effort for training machine learning models.

Active learning methods aim to improve data efficiency by querying the labels of samples presumed informative, in a feedback-driven fashion. In recent years, the pool-based active learning setting, in which queries are drawn from a large, static pool of unlabeled samples has drawn significant attention [7, 8, 9]. This is due to the abundance of such unlabeled pools and the compatibility of the pool-based setting with the training of deep neural networks.

Most active learners rely on defining a notion of *informativeness* of a given sample, such as model uncertainty [10, 11], expected model change [12, 13] or expected error reduction [14, 15]. However, in the batch setting, where multiple samples are queried simultaneously, not contemplating the information overlap between the samples can lead to sub-optimal queries. Consequently, batch *diversity* needs to be taken into account, often at the expense of individual sample informativeness [16].

In this paper, we demonstrate how a *constrained learning* formulation of the problem enables the use of *Lagrangian duality* for detecting informative samples. In particular, we bound the loss incurred by

---

*Corresponding author.

36th Conference on Neural Information Processing Systems (NeurIPS 2022).

each sample, and use the dual variables associated to these constraints as a measure of informativeness. We show that dual variables are directly related to the variations of the average optimal loss over the entire data distribution, which motivates our approach.

Through an iterative primal-dual strategy, we optimize the model parameters as well as the dual variables. We then leverage the learned embedding space [17, 18] to train a *dual regression head* that estimates the dual variable associated to each unlabeled sample. Our proposed active learning approach, which we refer to as Active Learning via Lagrangian dualitY, or ALLY can then be used to select a diverse and informative set of samples. We also argue that, under certain conditions, the strategy put forward is not severely impacted by the distribution shift induced by active sampling.

We evaluate the performance of ALLY on a suite of classification and regression tasks, and show that it performs similar to or better than state-of-the-art batch active learning methods. We further demonstrate how the trained backbone, alongside the dual regression head, enable the generation of novel samples that can be optimized to be maximally informative, shedding light on the interpretability of the proposed active learning framework.

## 2    Related Work

### 2.1    Active Learning

The literature on active learning is voluminous and a myriad of strategies for the pool-based setting have been proposed [7]. In what follows, we describe some of the approaches most connected to our work.

A simple way of measuring model uncertainty is by computing the entropy of the predicted class distribution. Designed for the sequential case, Entropy Sampling [10] selects the unlabeled sample with highest associated output entropy. Among the relevant methods is BADGE [15], which employs a lower bound on the norm of the gradients in the final layer of the network as a measure of informativeness. BADGE balances diversity and informativeness by using the $k$-MEANS++ seeding algorithm to select a batch with large Gram determinant in the gradient space. BAIT [19] builds on traditional, Information Matrix based methods to efficiently select a batch that optimizes a bound on the MLE error in a two stage manner. Other methods that propose notions of informativeness are BALD [20], which uses the mutual information between predictions and model parameters as an uncertainty measure, and Learning Loss [21], which trains a loss prediction module and queries the samples that hypothetically generate high errors (and, thus, large model updates). VAAL [22] and WAAL [23] are methodologically similar to [21] in the sense that they all use a multipartite system consisting of a feature encoder, a prediction head and an auxiliary estimator. This is also the case for the strategy presented in this paper.

Some diversity-promoting approaches are compatible with many informativeness measures. A popular approach is to cluster the samples of the unlabeled set and then select informative points from each cluster [24, 25, 26]. In [27], Monte Carlo sampling is used to simulate sequences of length $b$ of the sequential algorithm, and then a *best-matching* combination of the sequences is used to build a batch. A simpler approach is to select the $b$ most informative points after a stochastic perturbation of the informativeness scores [28].

Some methods do not enforce informativeness explicitly, but rather query a set of data points that is maximally representative of the entire unlabeled set. Coreset [8], for instance, formulates pool-based active learning as a core-set selection problem, and aims to identify a set of points that geometrically covers the entire representation space. To do this, Coreset selects the batch that when added to the labeled set, minimizes the maximum distance between labeled and unlabeled examples. Coreset is compatible with deep neural networks and can be used in both regression and classification settings. Similarly, DAL [29] emphasizes representativeness by framing active learning as a binary classification task and selecting queries that maximize the similarity between the labeled and unlabeled set.

### 2.2    Constrained Learning

The need to tailor the behavior of machine learning systems has led to the development of a constrained learning theory. A common approach is to use regularization, that is, to modify the learning objective

so as to promote certain requirements. However, choosing the level of regularization through a dimensionless weight can be more challenging than setting a constraint level. Furthermore, recent works [30, 31, 32] show that, from a PAC (Probably Approximately Correct) perspective, learning under requirements is as hard as classical learning and that it can be done in practice through primal-dual learners. This has led to numerous applications across several areas of ML such as federated learning [33], fairness [30], stability of neural networks [34, 35], adversarial robustness [36] and data augmentation [37].

# 3 Problem Formulation

## 3.1 Batch Active Learning

Let $\mathfrak{D}$ denote a probability distribution over data pairs $(\boldsymbol{x}, y)$, where $\boldsymbol{x} \in \mathcal{X} \subseteq \mathbb{R}^D$ represents a feature vector (e.g., the pixels of an image) and $y \in \mathcal{Y} \subseteq \mathbb{R}$ represents a label or measurement. In classification tasks, $\mathcal{Y}$ is a subset of $\mathbb{N}$, whereas in regression, $\mathcal{Y} = \mathbb{R}$.

Initially, a set $\mathcal{L} = \{(\boldsymbol{x}_i, y_i)\}_{i \in \mathcal{N}_{\mathcal{L}}}$ of data pairs, or labeled samples, is available, coming from a probability distribution $\mathfrak{D}$. This set is used to learn a predictor $f(\mathbf{x}; \mathcal{L}) : \mathcal{X} \to \mathcal{Y}$ from a hypothesis class $\mathcal{F}$. Note that dependence on the set used to learn $f$ is made explicit. Then, a batch $\mathcal{B}$ of samples, or *queries*, is selected from a pool of unlabeled samples $\mathcal{U} = \{\boldsymbol{x}_i\}_{i \in \mathcal{N}_{\mathcal{U}}}$ and sent to an oracle for labeling. The goal is to the select the batch that minimizes the expected loss over the natural data distribution. More precisely, we formulate the Batch Active Learning (BAL) problem as

$$\mathcal{B}^\star = \underset{\mathcal{B} \subseteq \mathcal{U} \,:\, |\mathcal{B}| \leq b}{\arg\min} \ \underset{f \in \mathcal{F}}{\min} \mathbb{E}_{(\mathbf{x}, y) \sim \mathfrak{D}} \left[ \ell \left( f(\mathbf{x}; \mathcal{L} \cup \mathcal{B}), y \right) \right], \tag{BAL}$$

where $b$, referred to as the *budget*, represents the maximum cardinality of $\mathcal{B}$ and $\ell : \mathcal{Y} \times \mathcal{Y} \to \mathbb{R}$ is a loss function (e.g., cross-entropy loss or mean-squared error).

This process is typically repeated multiple times. At each iteration, two main steps are performed: (i) selecting $\mathcal{B}$ and updating the sets: $\mathcal{L}^{(t)} = \mathcal{L}^{(t-1)} \cup \mathcal{B}$ and $\mathcal{U}^{(t)} = \mathcal{U}^{(t-1)} \setminus \mathcal{B}$, and (ii) obtaining the predictor $f$ with the aggregate set of labeled samples $\mathcal{L}^{(t)}$. Steps (i) and (ii) correspond to the outer and inner minimization problems in (BAL), respectively. In what follows, we focus on a single iteration, and thus obviate the dependence on the iteration $t$ and the set used to ease the notation.

## 3.2 Constrained Statistical Learning

Most active learning methods in the literature [8, 15, 20, 5, 7] formulate step (ii) above as an *unconstrained* Statistical Risk Minimization (SRM) problem [38],

$$\underset{f \in \mathcal{F}}{\min} \ \mathbb{E}_{(\boldsymbol{x}, y) \sim \mathfrak{D}} \left[ \ell \left( f(\boldsymbol{x}), y \right) \right]. \tag{SRM}$$

Our approach, alternatively, uses a *Constrained* Statistical Learning (CSL) formulation,

$$P^\star = \underset{f \in \mathcal{F}}{\min} \quad \mathbb{E}_{(\boldsymbol{x}, y) \sim \mathfrak{D}} \left[ \ell \left( f(\boldsymbol{x}), y \right) \right] \tag{CSL-a}$$

$$\text{s.t.} \quad \ell' \left( f(\boldsymbol{x}), y \right) \leq \epsilon(\boldsymbol{x}), \quad \mathfrak{D}_{\boldsymbol{x}}\text{–a.e.} \tag{CSL-b}$$

where $\ell' : \mathcal{Y} \times \mathcal{Y} \to \mathbb{R}$ is a secondary loss function, $\epsilon : \mathcal{X} \to \mathbb{R}$ is a mapping from each data point to a corresponding constraint upper bound, and $\mathfrak{D}_{\boldsymbol{x}}$ denotes the marginal distribution over $\mathcal{X}$. Note that the objective function in (CSL-a) is the same as in (SRM), but the secondary loss is required to be bounded $\mathfrak{D}_{\boldsymbol{x}}$–almost everywhere.

Letting $\lambda : \mathcal{X} \to \mathbb{R}^+$ denote the dual variable function, the Lagrangian associated to (CSL) can be written as

$$L(f, \lambda) = \mathbb{E}_{(\boldsymbol{x}, y) \sim \mathfrak{D}} \left[ \ell(f(\boldsymbol{x}), y) \right] \quad + \int_{\mathcal{X}, \mathcal{Y}} \lambda(\boldsymbol{x})(\ell'(f(\boldsymbol{x}), y) - \epsilon(\boldsymbol{x})) p(\boldsymbol{x}, y) d\boldsymbol{x} dy$$

$$= \mathbb{E}_{(\boldsymbol{x}, y) \sim \mathfrak{D}} \left[ \ell(f(\boldsymbol{x}), y) + \lambda(\boldsymbol{x})(\ell'(f(\boldsymbol{x}), y) - \epsilon(\boldsymbol{x})) \right],$$

where it is implicitly assumed that the conditional distribution $p(y|\boldsymbol{x})$ is a Dirac delta distribution, i.e., $y$ is a deterministic function of $\boldsymbol{x}$. This leads to the dual problem,

$$D^\star = \underset{\lambda \in \Lambda}{\max} \ \underset{f \in \mathcal{F}}{\min} \ L \left( f, \lambda(\boldsymbol{x}) \right), \tag{D-CSL}$$

where $\Lambda := \{\lambda \,|\, \lambda(\boldsymbol{x}) \geq 0, \ \mathfrak{D}_{\boldsymbol{x}}\text{–a.e.}\}$. There are three main motivations for this infinite programming formulation:

1. **Access to variations of $P^\star$**: As we show in Theorem 3.2, this formulation gives us access to $\partial P^\star(\epsilon(\boldsymbol{x}))$, enabling the use of *dual variables* as an indicator of the *informativeness* of the training samples.

2. **Considering both average and worst-case losses**: The most informative samples often lie in the tails of the distribution $\mathfrak{D}$. Those samples appear less frequently in the dataset and thus, models obtained by solving (SRM) can achieve low *average* errors without learning to classify/regress them correctly. Our formulation bridges the gap between the (SRM) and a worst-case Feasibility Statistical Learning (FSL) formulation,

$$P^\star = \min_{f \in \mathcal{F}} \quad 0 \tag{FSL-a}$$

$$\text{s.t.} \quad \ell\,(f(\boldsymbol{x}), y) \leq \epsilon(\boldsymbol{x}), \quad \mathfrak{D}_{\boldsymbol{x}}\text{–a.e.} \tag{FSL-b}$$

   which considers the worst-case loss.

3. **Adaptive regularization**: For a fixed dual variable $\lambda$, the Lagrangian is a regularized objective, where $\ell'$ acts as a regularizing functional. Thus, the max-min formulation in D-CSL can be viewed as a regularized minimization, where the regularization weight is updated during the training procedure according to the degree of constraint satisfaction or violation.

The dual problem can be interpreted as finding the tightest lower bound on $P^\star$. In the general case, $D^\star \leq P^\star$, which is known as weak duality. Nevertheless, under certain conditions, $D^\star$ attains $P^\star$ (strong duality) and we can derive a relation between the solution of (D-CSL) and the sensitivity of $P^\star$ with respect to $\epsilon(\boldsymbol{x})$. See Appendix **??** for more details.

In the following, we define the *Fréchet subdifferential* of a convex function, which allows us to justify the use of dual variables as a measure of *informativeness* of a sample.

**Definition 3.1.** Let $U, V$ be Banach spaces. The Fréchet subdifferential of a functional $P : U \to V$ at $u \in U$ is defined as:

$$\partial P(u) = \{z \in U^* \,:\, P(v) - P(u) \geq \langle z\,,\, v - u \rangle \text{ for all } v \in U\},$$

where $U^*$ denotes the topological dual space of $U$, and $\langle z\,,\, v - u \rangle = \mathbb{E}_{\mathfrak{D}}\left[z(\boldsymbol{x})(v(\boldsymbol{x}) - u(\boldsymbol{x}))\right]$.

Having the above definition, we state following theorem, which characterizes the variations of $P^\star$ (the optimum value of CSL) as a function of the constraint tightness $\epsilon(\boldsymbol{x})$.

**Theorem 3.2.** *If the problem* (CSL) *is strongly dual, then for any $\boldsymbol{x} \in \mathcal{X}$, we have*

$$-\lambda^\star(\boldsymbol{x}) \,\in\, \partial P^\star(\epsilon(\boldsymbol{x})),$$

*where $\partial P^\star(\epsilon(\boldsymbol{x}))$ denotes the Fréchet subdifferential of $P^\star$ with respect to $\epsilon(\boldsymbol{x})$, and $\lambda^\star(\boldsymbol{x})$ is the optimal dual variable associated to the constraint on $\boldsymbol{x}$.*

*Proof.* See Appendix **??**. $\qquad\square$

For any $\boldsymbol{x_0} \in \mathcal{X}$, let $\delta_{\boldsymbol{x_0}}(\boldsymbol{x})$ be a bump function of radius $r > 0$, centered at $\boldsymbol{x_0}$ (i.e., a continuous, radially-increasing function with support in $\|\boldsymbol{x} - \boldsymbol{x_0}\| \leq r$). Theorem 3.2 implies that

$$P^\star(\epsilon(\boldsymbol{x}) + \delta_{\boldsymbol{x_0}}(\boldsymbol{x})) - P^\star(\epsilon(\boldsymbol{x})) \geq \langle\, -\lambda^\star(\boldsymbol{x_0}),\, \delta_{\boldsymbol{x_0}}(\boldsymbol{x})\, \rangle.$$

The problem (CSL) typically includes an infinite number of constraints. Theorem 3.2 implies that the constraint whose perturbation has the most *potential impact* on the optimal value of (CSL) is the constraint with the highest associated optimal dual variable. For instance, infinitesimally tightening the constraint in a neighbourhood $\boldsymbol{x_0}$ would restrict the feasible set, causing an increase of the optimal value of (CSL) at a rate larger than $\lambda^\star(\boldsymbol{x_0})$. In that sense, the magnitude of the dual variables can be used as a measure of informativeness. Similarly to non-support vectors in SVMs [39], samples associated to inactive constraints (i.e., $\{\boldsymbol{x_0} : \lambda^\star(\boldsymbol{x_0}) = 0\}$), are considered uninformative.

## 3.3 On the Statistical Bias Induced by Active Sampling

In pool-based active learning, the resulting training set may not be representative of the natural data distribution $\mathfrak{D}$ [40, 41, 42]. This is due to the fact that queried samples are not randomly selected and often lie in the tails of $\mathfrak{D}$. Therefore, when performing several active learning iterations, we undertake a biased version of (BAL):

$$\mathcal{B}^\star = \underset{\mathcal{B} \subseteq \mathcal{U}^{(t)} \,:\, |\mathcal{B}| \leq b}{\arg\min} \; \min_{f \in \mathcal{F}} \mathbb{E}_{(\mathbf{x}, y) \sim \mathfrak{A}^{(t)}} \left[ \ell \left( f(\mathbf{x}; \mathcal{L}^{(t)} \cup \mathcal{B}), y \right) \right], \tag{bBAL}$$

where $\mathfrak{A}^{(t)}$ represents the biased distribution underlying the actively sampled set $\mathcal{L}^{(t)}$. Even though, at each iteration, the queried samples have some desired property (e.g., large impact on the expected loss), the learned predictor $f(\mathbf{x}; \mathcal{L}^{(t)} \cup \mathcal{B})$ is usually sub-optimal for the natural data distribution.

One way of undertaking the inner minimization in (bBAL) is with the worst-case/feasibility formulation,

$$P^\star = \min_{f \in \mathcal{F}} \quad 0 \tag{bFSL-a}$$

$$\text{s.t.} \quad \ell\left( f(\boldsymbol{x}), y \right) \leq \epsilon(\boldsymbol{x}), \quad \mathfrak{A}^t_{\boldsymbol{x}}\text{–a.e.} \tag{bFSL-b}$$

This formulation is closely related to the constrained formulation in (CSL). In Appendix **??**, we relate the dual problem of (bFSL) with (D-CSL) to argue that the impact of the inconsistency between distributions on our method is small. The key observation is that, if the marginal distributions $\mathfrak{D}_{\mathbf{x}}$ and $\mathfrak{A}^{(t)}_{\mathbf{x}}$ have the same support, then the respective feasibility formulations are equivalent, regardless of the probability distribution.

# 4 Proposed Approach

In light of the results mentioned in Section 3.2 on the usefulness of dual variables in constrained statistical learning as a measure of sample informativeness, we present our proposed method, ALLY, in this section. We start by introducing a primal-dual procedure to empirically solve the constrained learning problem, and we will then proceed to describe our active learning algorithm in detail.

## 4.1 Constrained Empirical Risk Minimization

The formulation in (D-CSL) poses two challenges: (i) the distribution $\mathfrak{D}$ is usually unknown and (ii) it is an infinite-dimensional problem since it optimizes over the functional spaces $\mathcal{F}$ and $\Lambda$. We handle the former by replacing expectations by their sample means over a set of labeled samples $\mathcal{L} = \{(\boldsymbol{x}_i, y_i)\}_{i \in \mathcal{N}_{\mathcal{L}}}$, as described in the classical Empirical Risk Minimization (ERM) theory [38, 43]. In order to resolve the latter, we introduce a *parameterization* of the hypothesis class $\mathcal{F}$ as $\mathcal{P} = \{f_{\boldsymbol{\theta}} \,|\, \boldsymbol{\theta} \in \Theta\}$, while we create a separate dual variable $\lambda_i$ for each training sample $\boldsymbol{x}_i$. These modifications lead to the Constrained Empirical Risk Minimization (CERM) problem,

$$\hat{P}^\star = \min_{\boldsymbol{\theta} \in \Theta} \quad \frac{1}{|\mathcal{N}_{\mathcal{L}}|} \sum_{i \in \mathcal{N}_{\mathcal{L}}} \ell\left( f_{\boldsymbol{\theta}}(\boldsymbol{x}_i), y_i \right) \tag{CERM-a}$$

$$\text{s.t.} \quad \ell'\left( f_{\boldsymbol{\theta}}(\boldsymbol{x}_i), y_i \right) \leq \epsilon_i, \; \forall i \in \mathcal{N}_{\mathcal{L}}. \tag{CERM-b}$$

This, in turn, results in the corresponding empirical dual problem,

$$\hat{D}^\star = \max_{\boldsymbol{\lambda} \geq \mathbf{0}} \; \min_{\boldsymbol{\theta} \in \Theta} \hat{L}(\boldsymbol{\theta}, \boldsymbol{\lambda}), \tag{D-CERM}$$

where $\boldsymbol{\lambda} = \{\lambda_i\}_{i \in \mathcal{N}_{\mathcal{L}}}$, $\boldsymbol{\lambda} \geq \mathbf{0}$ represents element-wise non-negativity, $\epsilon_i$ denotes the constraint upper bound associated to the $i^{\text{th}}$ point-wise constraint, and the *empirical* Lagrangian, $\hat{L}(\boldsymbol{\theta}, \boldsymbol{\lambda})$, is defined as

$$\hat{L}(\boldsymbol{\theta}, \boldsymbol{\lambda}) = \frac{1}{|\mathcal{N}_{\mathcal{L}}|} \sum_{i \in \mathcal{N}_{\mathcal{L}}} \left[ \ell(f_{\boldsymbol{\theta}}(\boldsymbol{x}_i), y_i) + \lambda_i \left[ \ell'(f_{\boldsymbol{\theta}}(\boldsymbol{x}_i), y_i) - \epsilon_i \right) \right] \right].$$

The max-min problem (D-CERM) can be undertaken by alternating the minimization with respect to $\boldsymbol{\theta}$ and the maximization with respect to $\boldsymbol{\lambda}$ [44, 30, 32], which leads to the primal-dual constrained

---

**Algorithm 1** Primal-dual constrained learning (PDCL)

---

1: **Input:** Labeled dataset $\mathcal{L}$, primal learning rate $\eta_p$, dual learning rate $\eta_d$, number of iterations $T$, number of primal steps per iteration $T_p$, constraint vector $\boldsymbol{\epsilon}$.
2: Initialize: $\boldsymbol{\theta}, \boldsymbol{\lambda} \leftarrow \mathbf{0}$.
3: **for** $t = 1, \dots, T$ **do**
4: $\quad \boldsymbol{\theta} \leftarrow \boldsymbol{\theta} - \eta_p \nabla_{\boldsymbol{\theta}} \hat{L}(\boldsymbol{\theta}, \boldsymbol{\lambda}) \quad (\times T_p)$ $\qquad$ // Update primal variables ($T_p$ SGD steps)
5: $\quad s_i \leftarrow \ell'\left(f_{\boldsymbol{\theta}}\left(\boldsymbol{x}_i\right), y_i\right) - \epsilon_i, \ \forall i \in \mathcal{N}_{\mathcal{L}}.$ $\qquad$ // Evaluate constraint slacks
6: $\quad \lambda_i \leftarrow \left[\lambda_i + \eta_d s_i\right]_+, \ \forall i \in \mathcal{N}_{\mathcal{L}}.$ $\qquad$ // Update dual variables
7: **end for**
8: **Return:** $\boldsymbol{\theta}, \boldsymbol{\lambda}$.

---

---

**Algorithm 2** Active learning via Lagrangian duality (ALLY)

---

1: **Input:** Labeled set $\mathcal{L}$, unlabeled set $\mathcal{U}$, primal learning rate $\eta_p$, dual learning rate $\eta_d$, number of PDCL iterations $T$, number of primal steps per iteration $T_p$, constraint vector $\boldsymbol{\epsilon}$.
2: $\boldsymbol{\theta}^\star, \boldsymbol{\lambda}^\star \leftarrow \text{PDCL}(\mathcal{L}, \eta_p, \eta_d, T, T_p, \boldsymbol{\epsilon}).$ $\qquad$ // Run the Primal-Dual Algorithm
3: $\boldsymbol{\omega}^\star \leftarrow \arg\min_{\boldsymbol{\omega}} \frac{1}{|\mathcal{N}_{\mathcal{L}}|} \sum_{i \in \mathcal{N}_{\mathcal{L}}} \|f_{\boldsymbol{\omega}}(f_{\boldsymbol{\phi}^\star}(\boldsymbol{x}_i)) - \lambda_i^\star\|^2.$ $\quad$ // Train the dual regression head
4: $j^\star \leftarrow \arg\max_{j \in \mathcal{N}_{\mathcal{U}}} f_{\boldsymbol{\omega}^\star}(f_{\boldsymbol{\phi}^\star}(\boldsymbol{x}_j))$ $\qquad$ // Find sample with highest dual variable
5: **Return:** $j^\star$.

---

learning procedure in Algorithm 1. Notice that $\min_{\boldsymbol{\theta} \in \Theta} \hat{L}(\boldsymbol{\theta}, \boldsymbol{\lambda})$ is the minimum of a family of affine functions on $\boldsymbol{\lambda}$, and thus is concave. Consequently, the outer problem corresponds to the maximization of a concave function and can be solved via gradient ascent. The inner minimization, however, is generally non-convex, but there is empirical evidence that deep neural networks can attain *good* local minima when trained with stochastic gradient descent [45, 46]. Some theoretical remarks on Algorithm 1 can be found in Appendix **??**.

As shown in Algorithm 1, the dual variables accumulate the slacks (i.e., distances between the per-sample secondary loss and constraint values) over the entire learning procedure. This allows the dual variables to be used as a measure of informativeness, while at the same time affecting the local optimum to which the algorithm converges. Quite interestingly, similar ideas on monitoring the evolution of the loss for specific training samples in order to recognize impactful instances have been used in several generalization analyses [47, 48].

## 4.2 ALLY: Active Learning via Lagrangian Duality

Our proposed active learning algorithm, ALLY, is presented in Algorithm 2 (for the case of $b = 1$). Given a set of labeled samples, $\mathcal{L}$, we first obtain the model parameters $\boldsymbol{\theta}^\star$ and the dual variables associated to samples in $\mathcal{L}$ using the primal-dual constrained learning approach in Algorithm 1. Taking a representation learning approach [17, 18, 49, 50], we then partition the model $f_{\boldsymbol{\theta}^\star}$ to a *backbone* $f_{\boldsymbol{\phi}^\star} : \mathcal{X} \rightarrow \mathbb{R}^d$, where $d$ denotes the dimensionality of the *embedding space*, and a *prediction head* $f_{\boldsymbol{\psi}^\star} : \mathbb{R}^d \rightarrow \mathcal{Y}$, such that $f_{\boldsymbol{\theta}^\star} = f_{\boldsymbol{\phi}^\star} \circ f_{\boldsymbol{\psi}^\star}$ and $\boldsymbol{\theta}^\star = \boldsymbol{\phi}^\star \cup \boldsymbol{\psi}^\star$. In order to estimate the informativeness of the samples in the unlabeled dataset $\mathcal{U}$, we train a *dual regression head* $f_{\boldsymbol{\omega}} : \mathbb{R}^d \rightarrow \mathbb{R}^+$ on the embeddings generated by $f_{\boldsymbol{\phi}^\star}$ by minimizing the mean-squared error

$$L_{\boldsymbol{\lambda}}(\boldsymbol{\omega}) = \frac{1}{|\mathcal{N}_{\mathcal{L}}|} \sum_{i \in \mathcal{N}_{\mathcal{L}}} \|f_{\boldsymbol{\omega}}(f_{\boldsymbol{\phi}^\star}(\boldsymbol{x}_i)) - \lambda_i^\star\|^2, \tag{1}$$

while the parameters $\boldsymbol{\phi}^\star$, hence the embeddings, are kept frozen. It should be noted that the idea of mapping embeddings to dual variables is present in other machine learning settings [51]. Once the dual regression head is trained, we evaluate it on the embeddings corresponding to the unlabeled samples, and identify the sample with the highest predicted dual variable.

As explained in Section 2.1, in the batch setting, selecting the $b$ samples with the highest associated dual variables is not optimal, due to the potential information overlap of such samples [7, 16]. Our method is compatible with any batch diversity approach that takes informativeness scores and unlabeled data points (or embeddings) as inputs.

### 4.3 Connection to BADGE [15]

The BADGE method [15] uses the gradient of the loss function with respect to the parameters of the last layer, denoted by $\boldsymbol{\theta}_L$, as a measure of informativeness, i.e.,

$$\frac{\partial \ell(f_{\boldsymbol{\theta}}(\boldsymbol{x}), \hat{y}(\boldsymbol{x}))}{\partial \boldsymbol{\theta}_L}, \tag{2}$$

where $\hat{y}(\boldsymbol{x})$ is the *hypothetical* label of $\boldsymbol{x}$, defined as $\hat{y}(\boldsymbol{x}) := \arg\max_{y \in \mathcal{Y}} [f_{\boldsymbol{\theta}}(\boldsymbol{x})]_y$. In contrast, as discussed in Theorem 3.2, to evaluate the informativeness of a given sample, ALLY observes

$$\frac{\partial P^{\star}}{\partial \epsilon(\boldsymbol{x})} = \frac{\partial P^{\star}}{\partial \boldsymbol{\theta}} \frac{\partial \boldsymbol{\theta}}{\partial \epsilon(\boldsymbol{x})} = \frac{\partial \mathbb{E}_{(\boldsymbol{x},y) \sim \mathfrak{D}} \left[ \ell \left( f_{\boldsymbol{\theta}^{\star}}(\boldsymbol{x}), y \right) \right]}{\partial \boldsymbol{\theta}} \frac{\partial \boldsymbol{\theta}}{\partial \epsilon(\boldsymbol{x})}. \tag{3}$$

There are three main differences between these informativeness measures:

- ALLY uses the derivative of the average optimal loss *over the entire distribution*, whereas BADGE only considers the point-wise derivative. In fact, most strategies evaluate their scoring function, or informativeness measure, on a single sample. Note that the term $\frac{\partial \mathbb{E}_{(\boldsymbol{x},y) \sim \mathfrak{D}} [\ell(f_{\boldsymbol{\theta}^{\star}}(\boldsymbol{x}), y)]}{\partial \boldsymbol{\theta}}$ is constant for all $\mathbf{x}$.
- ALLY observes the gradient with respect to *all* model parameters, not only the ones in the last layer.
- Aside from the derivative of the loss with respect to the model parameters, ALLY also considers an additional term $\frac{\partial \boldsymbol{\theta}}{\partial \epsilon(\boldsymbol{x})}$, which models how the results of the optimization (i.e., model parameters) change when the constraint function is perturbed.

### 4.4 Interpreting the Informativeness Score of ALLY

Our proposed framework allows us to leverage the trained backbone and dual regression head in a *generative* manner to create novel samples that are most informative. Here, we focus on the MNIST dataset, and use the trained model to generate synthetic images with maximal associated dual variables. We begin by training a MLP on 10% of the MNIST dataset. Then, similarly to [52], we perform gradient ascent on images that are initially considered uninformative, so as to maximize their predicted dual variables. The progression of the resulting images is shown in Figure 1.

As the predicted dual variable increases, patterns corresponding to other digits appear, increasing the uncertainty on the true label of the image. For instance, the third column of Figure 1 shows a handwritten '1' that is progressively transformed into a blurred superposition of '7,' '2' and '1'. Images in the last row of Figure 1 can be interpreted as lying in the tails of the distribution $\mathfrak{D}$, or close to the decision boundary of the end-to-end model $f_{\boldsymbol{\theta}}$. This also suggests that informative samples and



Figure 1: Sample generation by maximization of predicted dual variables. The top row shows the initial images from MNIST, to which the predictor associates a low dual variable. Rows 2-5 display the images resulting from subsequent iterations of gradient ascent.

outliers (such as mislabeled samples) may be hard to distinguish. Recent empirical findings indicate that many active learning algorithms consistently prefer to acquire samples that traditional models fail to learn [53]. Thus, modifying ALLY in order to avoid sampling these so-called *collective outliers* (e.g., by setting an upper bound on the dual variable associated to the queries) would be desirable in datasets that are not highly curated.

# 5    Experimental Evaluation

## 5.1    Settings

We consider four image classification tasks and one biomedical, non-image regression task. In the classification setting, we use standard datasets that commonly appear in the active learning literature, namely STL-10 [54], CIFAR-10 [55], SVHN [56] and MNIST [57]. Lacking an established benchmark regression dataset for active learning, we evaluate ALLY on the Parkinsons Telemonitoring dataset (PTD) [58]. In this regression task, the goal is to predict UPDRS (Unified Parkinson's Disease Rating Scale) scores from dysphonia measurements such as variation in fundamental frequency. Since measurements in this dataset are the result of a *costly* clinical trial that requires expert knowledge, this task is a prime example in which active learning might be essential.

In all experiments, the initial labeled set $\mathcal{L}_0$ consists of 200 randomly drawn samples, and the budget is set to either $b = 200$ or $b = 1000$. We use a ResNet-18 architecture [59] with an embedding size of 128. In the case of MNIST and PTD, which are simpler tasks, we use a multi-layer perceptron (MLP) with two hidden layers, each with 256 neurons and rectified linear unit (ReLU) activation, leading to an embedding size of 256. The dual regression head $f_\omega$ is a MLP with 5 hidden layers, ReLU activations and batch normalization.

As done in [24, 25], to ensure diversity in the batch, we cluster the embeddings of the unlabeled samples, i.e., $\{f_{\phi^\star}(x_j)\}_{j \in \mathcal{N}_\mathcal{U}}$, using the $k$-MEANS clustering algorithm [60], where $k \leq b$ is a hyperparameter. We then select the samples with the highest associated dual variables from each cluster, while maintaining equity among the number of samples per cluster. As shown in Appendix **??**, the performance of ALLY improves with an increased number of clusters $k$, since it leads to a more diverse batch of selected samples. Therefore, in all our experiments, we set $k = b$.

Regarding the role of the secondary loss, we opted for a generic formulation as the performed sensitivity analysis in Theorem 3.2 holds for various choices of $\ell'$. However, in all our experiments, we set $\ell'(\cdot, \cdot) = \ell(\cdot, \cdot)$. We believe that using unsupervised or self-supervised losses for $\ell'$ is a promising research direction, and we leave it for future work.

We compare our algorithm with Entropy sampling, BADGE, Coreset and BAIT. While Entropy sampling focuses purely on uncertainty, BADGE and BAIT balance both diversity and informativeness with different approaches. Coreset differs from the previous methods in that it focuses on batch representativeness by framing active learning as a coreset selection problem. It has been observed that,

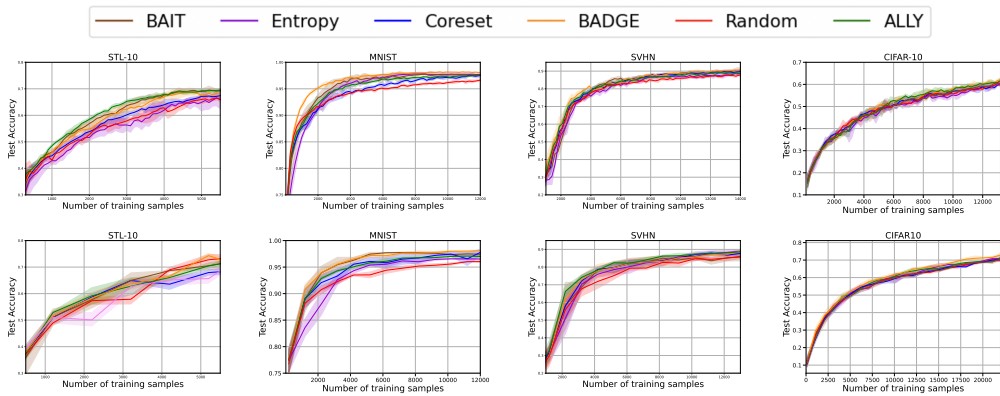

Figure 2: Accuracy in the test set as a function of the number of training samples in four classification settings and two budgets, 200 (top) and 1000 (bottom). Solid curves represent the mean across five different random seeds, while shaded regions correspond to the standard deviation.

in some scenarios, several active learning methods fail to consistently outperform Random Sampling [53, 61, 29]. We thus include it as one of the five baselines. As explained in section 4, ALLY uses a primal-dual approach to learn $f(\mathbf{x}; \mathcal{L} \cup \mathcal{U})$, while all other baselines are optimized using stochastic gradient descent (ADAM). We adopt the PyTorch [62] implementation of the baselines from [63].

## 5.2 Classification

The experiments on STL-10, CIFAR-10, SVHN and MNIST are all 10-class, image classification tasks. We use the cross-entropy loss for both $\ell(\cdot, \cdot)$ and $\ell'(\cdot, \cdot)$ and set $\epsilon(\boldsymbol{x}) = 0.2$, $\forall \boldsymbol{x}$. As shown in Figure 2, ALLY exhibits top-2 performance with respect to other baselines in STL-10 (budget 200), SVHN (budget 1000) and CIFAR-10 (budget 200). In these three datasets, the improvement in the number of samples needed by ALLY to achieve 97% of the final accuracy, in comparison with the best baseline, is 9%, 8% and 2%, respectively. In MNIST, however, BADGE and BAIT consistently outperform ALLY. This may be due to the fact that the MLP, being less expressive, yields embeddings of lower quality, hindering the prediction of dual variables.

## 5.3 Regression

We use mean-squared error for both $\ell(\cdot, y)$ and $\ell'(\cdot, y)$ and set $\epsilon(\boldsymbol{x}) = 0.1$, $\forall \boldsymbol{x}$. As seen in Figure 3, ALLY outperforms both Random and Coreset in this regression task, the gap being larger at the beginning of the learning curve. Note that BADGE and Entropy are not applicable, since they are limited to classification scenarios.

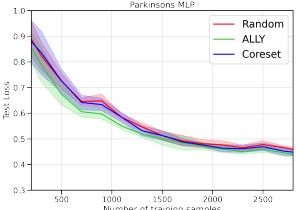

Figure 3: Mean-squared error in the test set as a function of the number of training samples in the Parkinson's telemonitoring dataset.

## 5.4 Ablation on the Constraint Tightness

Figure 4 illustrates the role $\epsilon$ plays in the optimization procedure. For extremely large values of epsilon (e.g., 20 nats), the constraint slacks become negative for all samples, and thus all dual variables become zero, making them uninformative (analogous to an unconstrained problem). Large values (e.g., 2 nats) can potentially be used to detect outliers. Our ablations suggest that values in the range $[1.05 p_u, 1.25 p_u]$, where $p_u$ is the average loss observed when training the model without constraints, work well in practice. The sensitivity of the method with respect to the constraint level is impacted both by the task/dataset at hand and the diversity technique used. It is reasonable to expect K-means to be less sensitive to the constraint level than stochastic perturbations of the informativeness scores.

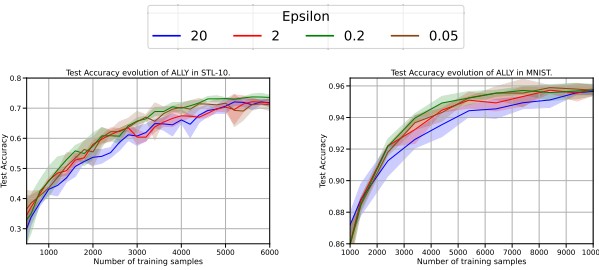

Figure 4: Ablation on the value of the constraint tightness $\epsilon$ on the performance of ALLY in STL-10 and MNIST.

## 5.5 Dataset Redundancy

We study the performance of ALLY and Entropy Sampling with varying levels of dataset redundancy in STL-10. We define the level of dataset redundancy as the number of copies of each sample present in the training set. We then analyze the evolution of test accuracy over several rounds of active learning with a query budget of 200.

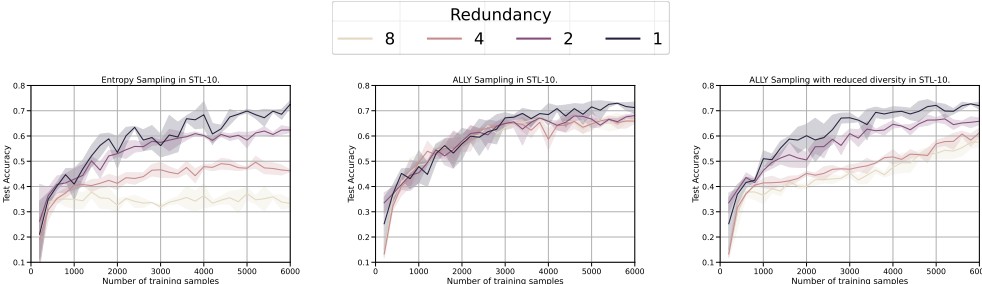

Figure 5: Impact of dataset redundancy on the performance of Entropy Sampling (left), ALLY (center) and low-diversity ALLY (right).

We observe that a strategy solely based on informativeness (e.g., Entropy Sampling) is very sensitive to this type of dataset redundancy. This can be attributed to the fact that, at each round, copies of the most informative samples are queried simultaneously, leading to low batch diversity. Moreover, ALLY appears to be more robust than Entropy Sampling to this type of dataset corruption. This is not surprising, since only one sample is queried from each cluster, avoiding batch information overlap. However, when increasing the number of samples queried from each batch, the performance of ALLY is degraded, resembling that of Entropy Sampling. This experiment suggests that the impact of dataset redundancy on active learning strategies is more linked to the diversity technique used than the informativeness measure. Whether this behaviour generalizes to other types of redundancy is left for future work.

## 6 Concluding Remarks

We presented ALLY, a principled batch active learning method based on Lagrangian duality. Our method formulates the learning problem using constrained optimization and solves it in the Lagrangian dual domain via a primal-dual approach. We leverage the fact that the magnitude of the dual variables can be viewed as a measure of informativeness of the corresponding training sample, as it indicates the sensibility of the optimum value of the objective function with respect to a perturbation in the respective constraint.

Following the completion of the primal-dual learning phase, we employ the learned sample representations, as well as their respective dual variables, to train a dual regression head. This predictor is used to estimate the dual variables associated to unlabeled samples. The resulting informativeness measure is compatible with several batch diversity promoting techniques. Using the k-MEANS algorithm, we demonstrated that this principled method exhibits competitive performance in several classification and regression experiments. We also showed that under certain conditions, the impact of the distribution shift induced by active sampling is small.

In our experiments, we have set the secondary loss to be identical to the primary supervised loss (i.e., cross-entropy loss for classification, and mean-squared error for regression). However, evaluating the performance of ALLY under alternative *unsupervised* or *self-supervised* secondary losses is a promising future direction. Finally, it would be interesting to evaluate the performance of ALLY under different diversity measures, comparing their computational burden.

## 7 Acknowldegments

Supported by NSF-Simons MoDL, Award #2031985, NSF AI Institutes program, Award #2112665, and NSF HDR TRipods Award #1934960.

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
