# OpenReview forum: "A Lagrangian Duality Approach to Active Learning"
_NeurIPS.cc/2022/Conference — NeurIPS 2022 Accept_

### Official Review · Reviewer_Jf6f · 2022-07-10

**Rating:** 5
**Confidence:** 3
**Soundness:** 2 fair
**Presentation:** 3 good
**Contribution:** 2 fair

**Summary:**

This paper proposes a new algorithm for batch-mode active learning, which is the problem of selecting and labeling a batch of unlabeled data points in order to yield the best possible improvement in a downstream model once trained. This algorithm uses Lagrangian duality in convex optimization to obtain an informativeness measure for every labeled data point; it then trains a model to predict the informativeness of unlabeled data and selects the most informative points from different clusters in the feature space. The algorithm demonstrates some improvement over classification and regression baselines.

**Questions:**

- Does strong duality actually hold in your empirical results? If not, how large is the duality gap? Moreover, how would a lack of strong duality affect the Frechet subdifferential result in Theorem 3.2?
- How does this method scale to and perform in larger data sets, e.g., ImageNet?
- The mathematical comparison vs BADGE is interesting. A useful ablation would be to compare the correlations between BADGE and ALLY embeddings. This is important because the theory doesn’t necessarily hold with non-convex losses or hypothesis classes.

**Limitations:**

Limitations are briefly discussed. Societal impact isn't discussed.

**Strengths And Weaknesses:**


**Strengths**

- Using Lagrangian dual variables as an informativeness measure is a novel idea. Moreover, the idea of estimating the dual variables for unlabeled data points presents an interesting approach to capture this informativeness.



**Weaknesses**

- The batch active learning problem (BAL) is written as selecting a set of points that minimizes the training set loss. However, this is incorrect and we typically want to minimize out-of-sample error/test loss (e.g., see Sener & Saverese ’18). Note that when we select what data points to train, we void any iid or distribution assumptions and thus, minimizing train loss does not guarantee low test loss.
- The methodology is heavily motivated by an assumption of strong duality, which holds only for convex loss functions and hypothesis sets.
- The idea of using 3 models, e.g., a feature encoder, a classifier head, and an auxiliary estimator, has been explored in several works, e.g., VAAL (Sinha et al ’19), WAAL (Shui et al ’20).


**References**

Sinha, Samarth, Sayna Ebrahimi, and Trevor Darrell. "Variational adversarial active learning." Proceedings of the IEEE/CVF International Conference on Computer Vision. 2019.

Shui, Changjian, et al. "Deep active learning: Unified and principled method for query and training." International Conference on Artificial Intelligence and Statistics. PMLR, 2020.

---

> ### Author Response · Authors · 2022-08-01
> **Response to Reviewer Jf6f**
>
> Thank you very much for your comments about our paper and for pointing out the novelty of our proposed method using Lagrangian duality. In what follows, we provide point-by-point responses to your questions and comments:
>
> - **Training Loss vs. Test Loss Minimization:** This is a great point, and we fully agree with the reviewer. Our ultimate goal is to minimize the loss over the natural data distribution, and we will change this in the camera-ready version of the paper.
> - **Strong Duality Assumption:** As mentioned by the reviewer, our results hinge on the assumption of strong duality, which needs convex losses and a strictly feasible point in the statistical domain. For non-convex losses, there exist some relevant bounds (see, e.g., Chamon, Luiz FO, Santiago Paternain, Miguel Calvo-Fullana, and Alejandro Ribeiro. "Constrained learning with non-convex losses." IEEE Transactions on Information Theory (2022)).
> - **Similar 3-model Architectures in Prior Work:** We agree that some methods in the active learning literature have similar architectures. Aside from the references you mentioned, another prior work with a similar architecture to ours is [21] (Donggeun Yoo and In So Kweon. Learning loss for active learning. CoRR, abs/1905.03677, 2019. URL http://arxiv.org/abs/1905.03677), in which an auxiliary module is trained to predict the loss. Nevertheless, please note that none of these methods is based on Lagrangian duality nor constrained optimization. We will discuss the aforementioned architectural similarities with prior work in the camera-ready version of the paper.
> - **Scaling to Larger Datasets:** We, unfortunately, do not have the computational capability to evaluate our proposed method (and the baselines) on ImageNet, but we will add experiments on ImageNette (a smaller version of ImageNet), as well as a larger-scale dataset, such as NWPU RESISC–45 (Cheng, Gong, Junwei Han, and Xiaoqiang Lu. "Remote sensing image scene classification: Benchmark and state of the art." Proceedings of the IEEE 105, no. 10 (2017): 1865-1883), to the camera-ready version of the paper.
> - **Ablation on ALLY and BADGE Embeddings:** Thank you very much for your suggestion. This is a very interesting experiment, and we will design and present an experiment on the comparison and correlation between the embeddings resulting from ALLY and BADGE in the camera-ready version of the paper.

---

> > ### Comment · Reviewer_Jf6f · 2022-08-08
> > **Thanks for the rebuttal**
> >
> > The convex assumption is a bit concerning since it almost never holds in practice. I would encourage the authors to include more discussion around this assumption.
> >
> > I would also encourage the authors to make the changes they promised with regards to scaling to larger data sets and ablating on ALLY vs BADGE embeddings.
> >
> > I have slightly increased my score under the assumption of the above changes.

---

> > > ### Author Response · Authors · 2022-08-09
> > > **Thank you!**
> > >
> > > Thank you very much for increasing your score. We will make sure to add more discussion on the convexity assumption, and we will also add the additional experiments to the camera-ready version of the paper.

---

### Official Review · Reviewer_GJBg · 2022-07-10

**Rating:** 7
**Confidence:** 3
**Soundness:** 4 excellent
**Presentation:** 4 excellent
**Contribution:** 3 good

**Summary:**

Batch active learning improves the sample qualities (avoid sample collapsing due to incremental improvement of the backbone model) and enables parallel oracles.  This paper tried to solve a critical issue for active learning. Unlike existing works, this paper proposed to solve the problem as a constrained learning problem and the algorithm solves the objective duality form.

**Questions:**

Continue with the cons of the pros and cons, it will be stronger if the authors could provide an empirical analysis of the query costs. For example, what is the wall-clock time of querying N examples to provide to the oracles? Practitioners of this method should have a clear understanding what is the overhead and when should use this proposed method.

**Strengths And Weaknesses:**

Pros:

The motivation is clear and well motivating.
The presentation is good and easy to follow.
The method proposed is interesting.

Cons:

Did not provide asymptotically analysis and empirical analysis about the query cost.

---

> ### Author Response · Authors · 2022-08-01
> **Response to Reviewer GJBg**
>
> Thank you very much for spending the time to review the paper, and thank you for your overall positive evaluation.
>
> With regards to your question about the query cost, we were unable to derive an asymptotic analysis of the query cost theoretically. However, in the camera-ready version of the paper, we will report the empirical wall-clock time of querying a set of $N$ samples using our proposed method. Please note that such wall-clock times will depend on the diversity method used; therefore, we will report the query run-times both with and without the $k$-means step.

---

> > ### Comment · Reviewer_GJBg · 2022-08-04
> > **Acknowledgement of response**
> >
> > Thanks for the detailed response from the authors. I appreciate the thorough response here.

---

> > > ### Author Response · Authors · 2022-08-09
> > > **Thank you!**
> > >
> > > Thank you very much for reading our response, and thanks again for your positive evaluation of our work.

---

### Official Review · Reviewer_v7Ec · 2022-07-13

**Rating:** 7
**Confidence:** 4
**Soundness:** 4 excellent
**Presentation:** 3 good
**Contribution:** 3 good

**Summary:**

The authors present a new active learning algorithm by formulating AL as a constrained statistical learning algorithm (CSL). In the CSL formulation the objective is to minimize loss on the data distribution subject to an auxiliary loss being upper bounded by epsilon everywhere. The authors then consider the Lagrangian of the optimization problem to derive a primal-dual algorithm to solve the optimization problem over the labeled dataset. The dual variables, obtained for the labeled dataset, are then generalized over the unlabeled pool by using a 2-head neural net, where one head predicts the labels and another head predicts the dual variables. This neural net is used to generalize the value of the dual variables to the entire unlabeled pool. The sample with the maximum dual variable is then chosen for labeling in the next round and this process is repeated. The authors provide an interpretation of the dual variables and also demonstrate the efficacy of their methods against a few baseline algorithms. The paper is well written and was easy to follow.

**Questions:**

1. What is the reason for formulating the AL problem as a CSL problem ? Can we reason about this choice of formulation from a statistical viewpoint?
2. The role of the auxiliary loss function was unclear to me? What would the advantage be of choosing an auxiliary loss function to be different from the main loss function?
3. Algorithm 2 chooses the unlabeled point with the maximum value of lambda as the data point to label next. Section 5.2 shows that lambda is a score of informativeness and large values of lambda are associated with very noisy samples. While I see wisdom in designing the algorithm to be noise seeking, would it be better to interpret lambda as probabilities and sample in proportion to lambda (favouring noisy samples in general) instead of choosing the sample with largest lambda? This would allow one to use importance weights in the CSL formulation and allow for optimizing an unbiased loss function.
4. While I understand the reason for the simplistic choice of epsilon ( epsilon(x) = c), can the authors suggest other possible choices? Would it be possible to design sensible heuristics or apply simple, unsupervised learning algorithms apriori to get good choice for epsilon and then use this choice in the algorithms? If yes, what would such heuristics or unsupervised algorithms look like?
5. The constant choice of epsilon and section 4.3 are at odds with each other. In Section 4.3, eqn 3, the authors take gradient w.r.t. epsilon. But, all experiments are with a constant epsilon. Furthermore,  it was not clear to me what the term \grad \theta/\grad epsilon that  appears in equation 3 captures? Could the authors shed some light here and interpret what this additional term do?
6. Are the networks retrained after addition of every single sample? If yes, are the old network values used to warm start training? A bit more clarity on the tricks used to train the network and compute optimizations would be useful.

**Ethics Review Area:**

["I don’t know"]

**Limitations:**

Computational limitations of repeatedly training the network were not addressed.

**Strengths And Weaknesses:**

Strengths:
The paper is well written and easy to follow.
The algorithm is pretty simple and intuitively sound.
The framing of active learning problem as a constrained statistical learning problem is innovative.
The experimental results look good.
On the whole I like the paper and I think it puts forward an interesting formulation and I can see quite a bit of follow up work stemming out of this work.

Weaknesses:
The motivation to frame AL as a CSL problem is not clear to me. In general the paper, while being good on the optimization front, does not do a good job in connecting the formulation to statistical principles.

Some design choices (such as choosing a constant epsilon) are not well explained.

---

> ### Author Response · Authors · 2022-08-01
> **Response to Reviewer v7Ec (Part 1)**
>
> Thank you very much for your time and effort in reviewing our paper and for your positive evaluation of our work. In what follows, we provide point-by-point responses to your questions:
>
> - **Motivation for the CSL Formulation of Active Learning:** As mentioned in Section 3.2, the unconstrained statistical risk minimization problem tries to minimize the *average* loss over the data distribution, i.e., $$\min_{f \in \mathcal{F}} \mathbb{E}_{(\boldsymbol{x}, y) \sim \mathfrak{D}}\left[\ell \left(f(\boldsymbol{x}), y\right)\right].$$ On the other hand, one can consider a *worst-case*/*feasibility* formulation, where per-sample losses are to be bounded instead of minimizing the average loss, i.e.,
> \begin{align*}
> \min_\{f \in \mathcal{F}} 0 ~~\text{s.t.} ~~ \ell \left(f(\boldsymbol{x}), y\right) \leq \epsilon(\boldsymbol{x}), \quad \mathfrak{D} \text {-a.e. }
> \end{align*} Our CSL formulation bridges the gap between the above two formulations, where both average and point-wise losses are considered simultaneously. In particular, please note that for the case where $\ell'=\ell$, the Lagrangian in Line 115 can be simplified as
> \begin{align*}
> L(f, \lambda)  =  \mathbb{E}\_{(\boldsymbol{x}, y) \sim \mathfrak{D}} \Big[(1+\lambda(\boldsymbol{x})) \ell (f(\boldsymbol{x}), y)  - \lambda(\boldsymbol{x}) \epsilon(\boldsymbol{x})\Big],
> \end{align*} which represents a *weighted* average loss, where the weights learn to adapt to the per-sample constraint satisfactions/violations (i.e., the dual variables). Therefore, we believe the CSL formulation is well-justified to be used in an active learning setting.
> - **Role of the Auxiliary Loss Function $\ell'$:** We included the secondary loss function $\ell'$ in the CSL problem formulation to make it more generic, in case it is desired to bound a different loss function almost everywhere than the primary loss function $\ell'$. This does not preclude the scenarios where $\ell'=\ell$, which is indeed what we use in all our experiments. However, as we discuss in the Concluding Remarks Section (Lines 299-302), the inclusion of the secondary loss function $\ell'$ allows us to potentially include unlabeled samples in the constraints as well by using unsupervised or self-supervised losses (e.g., contrastive losses). We believe selecting the best secondary loss function is, in and of itself, an exciting research question to investigate, which we leave as future work.

---

> ### Author Response · Authors · 2022-08-01
> **Response to Reviewer v7Ec (Part 2)**
>
> - **Treating Dual Variables as Probability Distributions:** This is a fantastic point, and we completely agree with the reviewer that the optimal dual variables can be treated as probability distributions that we can sample from to derive the most informative samples. Ideally, this would require a generative model (as noted in Section 5.2), which can create novel, informative samples that may not exist within the set of unlabeled samples. In the absence of such a generative model, we resorted to selecting the unlabeled samples with the highest estimated dual variables, alongside diversity considerations, for which we used $k$-means to add diversity to the selected batch. We also agree with your point that samples with extremely high dual variables might be "too noisy" or "outliers." The datasets that we used are very curated and exhibit very few of these outliers (such as mislabeled samples). In real-world datasets with noisier data, it would be reasonable to first clean the dataset or to set an upper bound on the dual variables of the queried samples. However, this might be challenging since informative samples (which we want to learn) and outliers/mislabeled samples (which we do not want to learn) might be hard to distinguish. As explained in Lines 294-297, many active learning methods have this problem (see, e.g., Karamcheti, Siddharth, Ranjay Krishna, Li Fei-Fei, and Christopher D. Manning. "Mind your outliers! investigating the negative impact of outliers on active learning for visual question answering." arXiv preprint arXiv:2107.02331 (2021).) We will add a discussion to the camera-ready version of the paper to better highlight this point.
> - **Choice of $\epsilon$:** As we noted in Section 5.5, we did use a reasonable heuristic method for setting the value of $\epsilon(\boldsymbol{x})$ as the average loss of the samples when the model is trained without any constraints, which makes the selected constant value for $\epsilon(\boldsymbol{x})$ both dataset- and model-dependent. We agree that more sophisticated unsupervised methods can also be used to set the value of $\epsilon$. As an example, one can make the bounds adaptive to each sample by formulating $\epsilon(\boldsymbol{x}) = c + \delta(\boldsymbol{x})$ with a small $c>0$, where $\delta(\boldsymbol{x})\geq 0$ represents an adaptive *slack* term that can adaptively relax the constraints if necessary. That said, such relaxations must be as small as possible for the problem formulation to remain meaningful. Therefore, a regularizing penalty term can be added to the objective, changing it to $\min_{f \in \mathcal{F}}  \mathbb{E}_{(\boldsymbol{x}, y) \sim \mathfrak{D}}\left[\ell \left(f(\boldsymbol{x}), y\right) + \mu \delta(\boldsymbol{x})\right]$, with $\mu>0$ being a hyperparameter. We will revise Section 5.5 in the camera-ready version of the paper to discuss alternative options for selecting $\epsilon(\boldsymbol{x})$.
> - **Gradients with respect to $\epsilon$:** Please note that although the value of $\epsilon$ can be constant for all samples, our theoretical analysis studies the impact of perturbing the value of $\epsilon(\boldsymbol{x})$ for a given sample $\boldsymbol{x}$. In particular, the terms $\frac{ \partial P^{\star} }{ \partial \epsilon(\boldsymbol{x})}$ and $\frac{ \partial \boldsymbol{\theta}}{ \partial \epsilon(\boldsymbol{x})}$ respectively represent the change in the optimal value and the set of model parameters if the value of $\epsilon(\boldsymbol{x})$ for a given sample $\boldsymbol{x}$ is slightly perturbed.
> - **Repeated Training and Warm-Starting:** As is commonplace in the active learning literature, for a fair comparison in this paper, we retrain the network from scratch after every query round, *without* warm-starting using the old parameter values. We agree that, in reality, such warm-starting could be beneficial to reducing the computational cost of ALLY (and any other active learning method), and we will mention that in the camera-ready version of the paper.

---

### Official Review · Reviewer_fegS · 2022-07-15

**Rating:** 7
**Confidence:** 3
**Soundness:** 3 good
**Presentation:** 4 excellent
**Contribution:** 3 good

**Summary:**

This paper tackles the pool-based active learning problem using a constrained learning formulation. The authors use the dual variable corresponding to each labeled data point as a measure of its information, and they train another model to predict the dual variable for the unlabeled data to determine which ones are the most informative, i.e., if the most informative ones are labeled and learned, the overall performance of the model will improve the most. The authors have also done experiments on classification tasks with images and a regression task with Unified Parkinson’s Disease Rating Scores. Additionally, there is a visualization of the relationship between the dual variable and the informativeness.

**Questions:**

###

1. The constrained optimization formulation can be think inversely where larger $\lambda$ corresponds to the data point that we do NOT want to include in learning. Here, we assume that including one data point in the learning process makes the model enjoy a lower loss on this specific data point. It is intuitive that having a stricter constraint will let the minimum of the objective function value becomes larger (i.e., worse) which means the overall loss is larger. Therefore, learning a data point with large $\lambda$ forces the model forgetting other information and the overall loss will only decrease more because of this learning. In this way, we want to learn the data with $\lambda=0$ because they do not hinder the minimization of the overall loss. Why is this explanation incorrect?
2. For the Batch Active Learning setting, Line 98, why the loss is evaluated on only the training dataset (already labeled + actively labeled)? I think we want the model to be good on any data from the population, not specifically on the ones we want to label.
3. We know that $\lambda$ depends on the derivative of $P$ in terms of $\epsilon$, but the derivative only captures the slope. Adding a newly labeled data point in model training may cause a large change of $\epsilon$. Is there any result on the second order derivative of $P$?
4. For Line 123, when you say ‘The most informative samples often lie in the tails of the distribution’, how do you define the informativeness strictly or specifically?
5. In the Abstract, Line 14, ‘(we) also discuss its limitations depending on the capacity of the model used’. Could you point out where the discussion is?

**Limitations:**

The authors have not discussed the limitation or the societal impact of their method.

**Strengths And Weaknesses:**

Major strengths:

1. This paper provides a new type of solution for active learning from a novel perspective of the usage of the dual variable in constrained optimization. Although Theorem 3.2 can be obtained straightforward with the Lagrangian function, it is very insightful to connect this formulation to the active learning problem where labeling and learning a new data point is explained as a way to satisfy a stricter constraint on the loss of this data point.
2. I would like to emphasize the merit of the simplicity of the formulation and analysis in this paper. Although the only requirement for Theorem 3.2, the strong dualality of CSL, is still strong, it leaves a space for readers to adapt the general idea in this paper to their own problem settings.

Minor strengths:

1. This paper is well-written.
2. The authors provide the ablation study for the constraint tightness.

Major weakness:

1. Theorem 3.2 could be wrong. The proof in Appendix B has some issues. For Line 577, $\lambda$ depends on $\epsilon$ but this is not shown. This affects the derivation in Line 580 where there is $\epsilon$. Also, the sub-gradient is not defined in the way as shown in Line 580. In general, this could be potentially fixed, but I still think it is a major problem because this is the crucial reason why one want to use the dual variable as the information indicator.

Minor weakness:

1. The citations in Line 87-88 are not correct.
2. The citations in the Appendix are missing, e.g., Line 558, 571-572, 585, etc.

---

> ### Author Response · Authors · 2022-08-01
> **Response to Reviewer fegS (Part 1)**
>
> Thank you very much for your extensive comments about our paper and for highlighting the novelty of our duality-based approach to active learning. In the following, we provide point-by-point answers to your questions and comments.
> - **Proof of Theorem 3.2:** We believe the proof is correct as we are considering the unperturbed problem with $\epsilon(\boldsymbol{x}) = 0$ and then using the fact that $f^\star$ is, by definition, feasible. However, we agree that the proof can be improved, and we will generalize it as follows:
> Define the Lagrangian $L(f, \lambda(\boldsymbol{x});\epsilon(\boldsymbol{x}))$ as $$L(f, \lambda(\boldsymbol{x});\epsilon(\boldsymbol{x})) =  \mathbb{E}_{(\boldsymbol{x}, y) \sim \mathfrak{D}} \Big[ \ell (f(\boldsymbol{x}), y)  + \lambda(\boldsymbol{x})( \ell'(f(\boldsymbol{x}), y)  - \epsilon(\boldsymbol{x}) )\Big],$$ where the dependence on $\epsilon(\boldsymbol{x})$ is explicitly shown. Then, following the definition of $P^\star(\epsilon(\boldsymbol{x}))$ in Line 577 and using strong duality, we have $$P^\star(\epsilon(\boldsymbol{x}))=\min_f L(f, \lambda^{\star}(\boldsymbol{x};\epsilon(\boldsymbol{x}));\epsilon(\boldsymbol{x})) \leq L(f, \lambda^{\star}(\boldsymbol{x};\epsilon(\boldsymbol{x}));\epsilon(\boldsymbol{x}))$$ with the inequality being true for any function $f$, and where the dependence of $\lambda^{\star}$ on $\epsilon(\boldsymbol{x})$ is also explicitly shown. Now, consider an arbitrary function $\epsilon'(\boldsymbol{x})$ and the respective primal function $f^{\star}(\cdot;\epsilon'(\boldsymbol{x}))$ which minimizes its corresponding Lagrangian. Plugging $f^{\star}(\cdot;\epsilon'(\boldsymbol{x}))$ into the above inequality, we have
> \begin{align*}
>     P^\star(\epsilon(\boldsymbol{x})) &\leq L(f^{\star}(\cdot;\epsilon'(\boldsymbol{x})), \lambda^{\star}(\boldsymbol{x};\epsilon(\boldsymbol{x}));\epsilon(\boldsymbol{x}))  =\mathbb{E}\_{(\boldsymbol{x}, y) \sim \mathfrak{D}} \Big[ \ell (f^{\star}(\boldsymbol{x};\epsilon'(\boldsymbol{x})), y)  + \lambda^{\star}(\boldsymbol{x};\epsilon(\boldsymbol{x}))( \ell'(f^{\star}(\boldsymbol{x};\epsilon'(\boldsymbol{x})), y)  - \epsilon(\boldsymbol{x}) )\Big].
>     \end{align*} Now, since $f^{\star}(\cdot;\epsilon'(\boldsymbol{x}))$ is *optimal* for constraint bounds given by $\epsilon'(\boldsymbol{x})$, we have $\mathbb{E}\_{(\boldsymbol{x}, y) \sim \mathfrak{D}} \: \Big[ \ell (f^{\star}(\boldsymbol{x};\epsilon'(\boldsymbol{x})), y)  \Big] = P^\star(\epsilon'(\boldsymbol{x}))$. Moreover, since $f^{\star}(\cdot;\epsilon'(\boldsymbol{x}))$ is *feasible* for constraint bounds given by $\epsilon'(\boldsymbol{x})$, we have $\ell'(f^{\star}(\boldsymbol{x};\epsilon'(\boldsymbol{x})), y)  \leq \epsilon'(\boldsymbol{x}), ~ \mathfrak{D}\_{\boldsymbol{x}} \text {-a.e. }$ Combining the above, we will get
> \begin{align*}
>     P^\star(\epsilon(\boldsymbol{x})) &\leq P^\star(\epsilon'(\boldsymbol{x})) + \mathbb{E}\_{(\boldsymbol{x}, y) \sim \mathfrak{D}} \Big[ \lambda^{\star}(\boldsymbol{x};\epsilon(\boldsymbol{x}))( \epsilon'(\boldsymbol{x})  - \epsilon(\boldsymbol{x}) )\Big]   = P^\star(\epsilon'(\boldsymbol{x})) + \langle \lambda^{\star}(\boldsymbol{x};\epsilon(\boldsymbol{x}))( \epsilon'(\boldsymbol{x})  - \epsilon(\boldsymbol{x}) )\rangle,
>     \end{align*} or equivalently, \begin{align*}
>      P^\star(\epsilon'(\boldsymbol{x})) - P^\star(\epsilon(\boldsymbol{x})) \geq \langle -\lambda^{\star}(\boldsymbol{x};\epsilon(\boldsymbol{x}))( \epsilon'(\boldsymbol{x})  - \epsilon(\boldsymbol{x}) )\rangle,
>     \end{align*} which exactly matches the definition of the Fréchet subdifferential in Definition 3.1, hence completing the proof. We will revise Appendix B to include the generalized proof.

---

> > ### Comment · Reviewer_fegS · 2022-08-02
> > **Still not quite clear or correct to me**
> >
> > Thanks for your detailed explanation. I think the writing here is much more clear and I actually have finished similar derivations by myself, but the problem I did not sort out still exists in your new proof.
> >
> > You consider
> > "an arbitrary function $\epsilon'(x)$ and the respective primal function $f^*(\cdot;\epsilon'(x))$ which minimizes its corresponding **Lagrangian**."
> >
> > (here I guess the Lagrangian means $L(f,\lambda(x);\epsilon(x))$ which include both $\ell(f(x),y)$ and $\lambda(x)(\ell'(f(x),y)-\epsilon(x))$)
> >
> >  and then you claim that $\mathbb{E}[\ell(f^*(x;\epsilon'(x)),y)]=P^*(\epsilon'(x))$ which confuses me since $P^*(\epsilon'(x))$ is defined as the Lagrangian value, not the objective function value.
> >
> > Therefore, I guess you implicitly used the fact that the optimal $f^*(\cdot;\epsilon'(x))$ satisfies the equality constraint $\ell'(f(x),y)-\epsilon(x)=0$ but I do not think it is always true since lower $\ell'(f(x),y)$. If it is not true, i.e., $\ell'(f(x),y) < \epsilon(x)$, the second last inequality in your response above will no longer hold.

---

> > > ### Author Response · Authors · 2022-08-02
> > > **Proof Clarification**
> > >
> > > Thank you for checking the updated proof, and this is a great question. To clarify, $f^{\star}(\cdot;\epsilon'(\boldsymbol{x}))$ is defined as
> > > \begin{align*}
> > > f^{\star}(\cdot;\epsilon'(\boldsymbol{x})) = \arg \min_f L(f, \lambda^{\star}(\boldsymbol{x};\epsilon'(\boldsymbol{x}));\epsilon'(\boldsymbol{x}))  = \arg \min_f \mathbb{E}\_{(\boldsymbol{x}, y) \sim \mathfrak{D}}  \Big[ \ell (f(\boldsymbol{x}), y)  + \lambda^{\star}(\boldsymbol{x};\epsilon'(\boldsymbol{x}))( \ell'(f(\boldsymbol{x}), y)  - \epsilon'(\boldsymbol{x}) )\Big]
> > > \end{align*} Note that this implies that the equality $\mathbb{E}\_{(\boldsymbol{x}, y) \sim \mathfrak{D}} \Big[ \ell (f^{\star}(\boldsymbol{x};\epsilon'(\boldsymbol{x})), y)  \Big] = P^\star(\epsilon'(\boldsymbol{x}))$ is indeed true since due to strong duality, complementary slackness holds, suggesting that $\mathbb{E}_{(\boldsymbol{x}, y) \sim \mathfrak{D}} \Big[ \lambda^{\star}(\boldsymbol{x};\epsilon'(\boldsymbol{x}))( \ell'(f^{\star}(\boldsymbol{x};\epsilon'(\boldsymbol{x})), y)  - \epsilon'(\boldsymbol{x}) )\Big]=0$. Please let us know if this explanation resolves your concerns, or if you have any further questions.

---

> > > > ### Comment · Reviewer_fegS · 2022-08-02
> > > > **Thanks for the response!**
> > > >
> > > > Using complementary slackness condition makes sense to me. I will change my rating to 'accept'.

---

> > > > > ### Author Response · Authors · 2022-08-02
> > > > > **Thank you!**
> > > > >
> > > > > We are happy that we were able to resolve your concerns about the proof, and we very much appreciate your prompt responses and your increased rating.

---

> ### Author Response · Authors · 2022-08-01
> **Response to Reviewer fegS (Part 2)**
>
> - **Incorrect and Missing Citations:** Thank you very much for pointing these out. Reference [30] is related to the stability of neural networks, while Reference [31] is related to federated learning. We will also fix the issue of missing references in the Appendix in the camera-ready version. Here are the missing references for your information:
>
>     [58] Adam M. Oberman and Jeff Calder. Lipschitz regularized deep neural networks converge and generalize. CoRR, abs/1808.09540, 2018. URL http://arxiv.org/abs/1808.09540.
>
>     [59] Dimitri Bertsekas, Angelia Nedic, and Asuman Ozdaglar. Convex analysis and optimization, volume 1. Athena Scientific, 2003.
>
>     [60] Alexander Shapiro. Semi-infinite programming, duality, discretization and optimality conditions. Optimization, 58(2):133–161, 2009. doi: 10.1080/02331930902730070. URL https://doi.org/10.1080/02331930902730070.
>
>     [61] J.Frédéric Bonnans and Alexander Shapiro. Perturbation analysis of optimization problems. Springer Science & Business Media, 2013. URL https://web.archive.org/web/20170809131322id_/http://www2.isye.gatech.edu/~ashapiro/publications/book-Jan00_typeset.pdf.
>
>     [62] Alexander Shapiro. Directional differentiability of the optimal value function in convex semi505 infinite programming. Math. Program., 70(1–3):149–157, Oct 1995. ISSN 0025-5610. doi:10.1007/BF01585933. URL https://doi.org/10.1007/BF01585933.
>
>     [63] R. Hettich P. Zencke. Directional derivatives for the value-function in semi-infinite programming. 38, 1987.
>
>     [64] Stephen Boyd and Lieven Vandenberghe. Convex Optimization. Cambridge University Press, March 2004. ISBN 0521833787.
>
>     [65] Kurt Hornik, Maxwell Stinchcombe, and Halbert White. Multilayer feedforward networks are universal approximators. Neural Networks, 2(5):359–366, 1989. ISSN 0893-6080. doi:https://doi.org/10.1016/0893-6080(89)90020-8. URL https://www.sciencedirect.com/science/article/pii/0893608089900208.
>
>     [66] Diederik P. Kingma and Jimmy Ba. Adam: A method for stochastic optimization. In Yoshua Bengio and Yann LeCun, editors, 3rd International Conference on Learning Representations, ICLR 2015, San Diego, CA, USA, May 7-9, 2015, Conference Track Proceedings, 2015. URL http://arxiv.org/abs/1412.6980

---

> > ### Comment · Reviewer_fegS · 2022-08-02
> > **Thanks for your response!**
> >
> > This reference list looks good to me.

---

> ### Author Response · Authors · 2022-08-01
> **Response to Reviewer fegS (Part 3)**
>
>  - **Sampling Data Points with Smaller Values $\lambda$:** This is a great point. You are correct that sampling data points with higher associated dual variables might make the model forget the already-learned samples. However, please note that the average loss objective would prevent that from happening, as it tries to minimize the loss over the *entire* data distribution (related to the comment below). On the other hand, sampling data points with smaller $\lambda$, which are already well-learned by the model, will not help the model learn any new information. This resembles the non-support vectors in support vector machines (SVMs): The samples with a zero dual variable are not support vectors, and including them does not affect the decision boundary, i.e., the model parameters, whereas only samples with positive dual variables (i.e., support vectors) impact the decision boundary.  Please also note that samples with *extremely* high dual variables might be "too noisy" or "outliers." The datasets that we used are very curated and exhibit very few of these outliers (such as mislabeled samples). In real-world datasets with noisier data, it would be reasonable to first clean the dataset or to set an upper bound on the dual variables of the queried samples. However, this might be challenging since informative samples (which we want to learn) and outliers/mislabeled samples (which we do not want to learn) might be hard to distinguish. As explained in Lines 294-297, many active learning methods have this problem (see, e.g., Karamcheti, Siddharth, Ranjay Krishna, Li Fei-Fei, and Christopher D. Manning. "Mind your outliers! investigating the negative impact of outliers on active learning for visual question answering." arXiv preprint arXiv:2107.02331 (2021)). We will add a discussion to the camera-ready version of the paper to better highlight this point.
>  - **Training vs. Population Loss:** We agree that the goal is to minimize the loss over the natural data distribution, but we end up doing something slightly different due to the sampling bias induced by our method (i.e., heavier tails). We will modify the formulation in the camera-ready version of the paper to reflect that.
>  - **Results on the second-order derivative of $P^{\star}$:** This is a very interesting question, but we are not aware of any such results in the literature, and it would be an exciting direction for future work.
>  - **Definition of Informativeness:** We define informativeness as the magnitude of the optimal dual variable associated with the sample, which highlights how difficult the sample is for the model to learn. We will revise Line 123 to better reflect our definition of informativeness of a given sample.
>  - **Limitations Due to Lower Model Capacity:** We briefly mention our method's limitations in Section 5.3 (Lines 265-267), where the performance of ALLY suffers with simpler backbone architectures, which might stem from their resultant lower-quality embeddings compared to their higher-capacity counterparts. We will add more experiments to the camera-ready version of the paper to better investigate the limitations of ALLY with respect to the backbone expressiveness.

---

> > ### Comment · Reviewer_fegS · 2022-08-02
> > **Thanks for your response!**
> >
> > The comment here clearly answers my questions.

---

### Meta-Review · Area_Chair_pKc4 · 2022-08-24

**Recommendation:** Accept
**Confidence:** Certain

**Metareview:**

In this paper, the authors formulate batch active learning as a constrained optimization problem, and develop a primal-dual approach to select a diverse set of unlabeled samples. The idea of using constrained optimization for active learning is novel and interesting, and the experimental results are also promising.

**Award:**

No

---

### Decision · Program_Chairs · 2022-09-14

Accept